# DualFocus: Integrating Macro and Micro Perspectives in Multi-modal Large Language Models

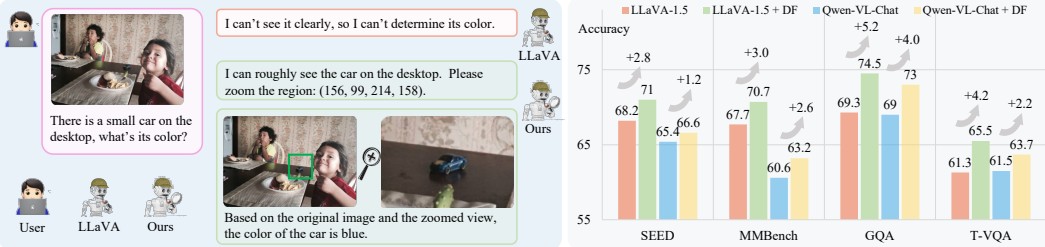

Figure 1: Demonstrating the efficacy of DualFocus (DF) in enhancing multi-modal large language model (MLLM) performance. The left panel illustrates the scenario where a user asks an MLLM to identify the color of a small car in an image. Unlike the baseline MLLM (LLaVA), which struggles with detail, the DualFocus approach integrates an auto-zoom operation that precisely localizes and enlarges the area of interest. Consequently, DualFocus can accurately discern and report the car's color. The right panel corroborates DualFocus's superior performance, presenting a clear advantage in accuracy across multiple benchmarks (SEED, MMBench, GQA, T-VQA) compared to the baseline models (LLaVA-1.5, Qwen-VL-Chat).

## Abstract

Current multi-modal large language models (MLLMs) predominantly focus on inputs from a global perspective, which results in deficiencies when addressing queries involving local regions. Drawing inspiration from human perceptual behavior, where zooming in on specific regions allows for more accurate inspection of fine details, this approach appears intuitive for improving model performance. However, effectively implementing this approach is challenging, primarily due to the diverse scenarios involved in localizing question-relevant regions, which can lead to potential errors. To address these challenges, we propose a novel solution, DualFocus, designed to enhance the model's ability to comprehend fine details while preserving its capacity for global contextual understanding. The DualFocus mechanism enables the model to first analyze an image from a macro (global) perspective and subsequently identify relevant sub-regions for focused micro-level analysis. By integrating the outputs from both macro and micro perspectives through a perplexity-guided selection process, the model can robustly address different tasks that may require either global context or detailed examination. Through comparative studies across different models and benchmarks, we demonstrate that DualFocus excels in balancing precise analysis with comprehensive understanding, significantly enhancing performance across a range of vision-language tasks.

## 1 Introduction

Large Language Models (LLMs) like ChatGPT (OpenAI, 2022), GPT-4 (OpenAI, 2023), and PaLM (Chowdhery et al., 2022) have revolutionized the field of natural language processing with their astounding ability to follow human instructions and tackle open-ended tasks. These models demonstrate an exceptional understanding of language and can generate text that is often indistinguishable from that produced by humans. Building upon this foundation, Multi-modal Large Language Models (MLLMs) such as MiniGPT-4 (Zhu et al., 2023), LLaVA (Liu et al., 2023b), and InstructBLIP

(Dai et al., 2023) have emerged, integrating the linguistic prowess of LLMs with visual understanding capabilities. Drawing on open-source LLMs like LLaMA (Touvron et al., 2023a), Qwen (Qwen, 2023), and InternLM (Team, 2023), these MLLMs extend their insight to the visual domain, allowing for a more comprehensive understanding of questions that necessitate both visual and textual processing.

One of the primary challenges in advancing MLLMs resides in effectively incorporating visual information. Early models such as MiniGPT-4 (Zhu et al., 2023) and LLaVA (Liu et al., 2023b) often rely on imagery of a fixed, small resolution. This approach simplifies processing but limits the model's ability to discern micro details crucial for answering specific questions.

Conversely, recent models such as Monkey (Li et al., 2023f), OtterHD (Li et al., 2023a), and LLaVA-NeXT (Liu et al., 2024) address fine-grained visual analysis by utilizing a high-resolution image divided into patches, supplemented by a low-resolution image for capturing global context. While this approach enhances the ability to analyze details, the substantial increase in image resolution introduces an overwhelming amount of information, much of which is irrelevant to the specific question at hand, making it more challenging to focus on useful information. Additionally, it incurs a quadratic growth in computational resource requirements as the input resolution increases.

Drawing inspiration from the human cognitive process, where individuals typically scan an image globally before focusing on specific details to answer a question, we propose a DualFocus strategy in MLLMs to imitate this behavior. The model first analyzes the entire image to capture the macro context, formulates the first answer from this global perspective, and then identifies key regions of interest. It then zooms into these identified subregions for a more detailed examination, enabling the second response to the given question. This approach extends the concept of the Chain of Thought (CoT) framework (Wei et al., 2022) by incorporating visual cues into the reasoning process through an automatic zoom mechanism.

Notably, during inference, the DualFocus model produces two potential answers: one from a macro perspective and another from a micro perspective. To effectively leverage both viewpoints and address potential inaccuracies in localizing question-relevant regions, we employ Perplexity (PPL) (Jelinek, 1998) as a decision metric. By comparing the losses associated with each answer, the model selects the one with the lower perplexity as the final prediction.

To equip MLLMs with the ability to localize question-relevant regions, we curated a new dataset derived from Visual Genome (VG) (Krishna et al., 2017), carefully selecting images and annotations to explicitly align with our DualFocus protocol. During training, the MLLM learns to identify relevant coordinates, define key subregions for a given query, and potentially encompass single or multiple related objects, thereby endowing the model with a robust "question-grounding" capability.

In our experiments, we utilize LLaVA 1.5 (Liu et al., 2023a) and Qwen-VL-Chat (Bai et al., 2023) as baseline models for their robust performance. Comparative experiments were conducted across model sizes of 7B and 13B parameters and a diverse set of benchmarks that ranged from multimodal and traditional VQA benchmarks. Specifically, DualFocus improves LLaVA 1.5 by 2.8, 3.0, 5.2, 4.2 and Qwen-VL-Chat by 1.2, 2.6, 4.0, 2.2, on SEED (Li et al., 2023c), MM-Benchmark (Liu et al., 2023c), TextVQA (Singh et al., 2019), and GQA (Hudson & Manning, 2019), respectively. Additionally, we observed a notable reduction in hallucinatory responses in MLLMs when tested on the POPE benchmark (Li et al., 2023e), highlighting the framework's potential to curb the generation of spurious detail by maintaining a balanced perspective. The comparative studies reinforce the versatility of DualFocus across a spectrum of benchmarks, affirming the effectiveness of the DualFocus mechanism.

## 2 RELATED WORK

### 2.1 LARGE LANGUAGE MODEL (LLM)

The evolution of LLMs has significantly shaped the natural language processing (NLP) landscape, showcasing the extraordinary capabilities of the transformer architecture. Initiated by encoder-decoder models such as BERT (Devlin et al., 2018), T5 (Raffel et al., 2020), and decoder-centric architectures like GPT (OpenAI, 2022), these models have excelled across various NLP tasks. With GPT3 (Brown et al., 2020), decoder-only models have become increasingly prevalent due to their

effectiveness in few-shot and zero-shot scenarios. Enhancements in model parameterization and dataset breadth are epitomized by Google's PaLM (Chowdhery et al., 2022), which pushed the performance boundaries of LLMs even further. To tailor models for natural conversational responses, strategies such as fine-tuning and reinforcement learning derived from human feedback have been adopted in InstructGPT (Ouyang et al., 2022) and ChatGPT (OpenAI, 2022). The open-source community has significantly contributed to ongoing innovation, with models such as (Touvron et al., 2023a), Vicuna (Chiang et al., 2023), Qwen (Qwen, 2023), LLaMA2 (Touvron et al., 2023b), Baichuan2 (Baichuan, 2023), and InternLM (Team, 2023).

## 2.2 MULTI-MODEL LARGE LANGUAGE MODEL (MLLM)

Recent research in MLLM has made significant advances. Different from previous works (Gupta & Kembhavi, 2023; Surís et al., 2023; Qi et al., 2024) in the visual programming area that leverages an LLM to call external programs to obtain visual knowledge, MLLMs explore the integration of visual knowledge into LLMs themselves. Models such as CLIP (Radford et al., 2021; Sun et al., 2023) and BLIP (Li et al., 2022) have demonstrated the effectiveness of contrastive learning to synchronize image and text modalities, remarkably improving zero-shot learning in tasks like Image Captioning and Image-Text Retrieval. Models such as MiniGPT-4 (Zhu et al., 2023), LLaVA (Liu et al., 2023b), InstructBLIP (Dai et al., 2023), and Otter (Li et al., 2023b) have pushed further, enhancing dialogic interactions and contextual understanding in image-text scenarios by focusing on precise pre-training alignments and fine-tuning processes. Notably, advanced techniques employing grounding data have been developed to anchor the models' perceptions more firmly in reality, as demonstrated by mPLUG-Owl (Ye et al., 2023), Shikra (Chen et al., 2023a), Opera (Huang et al., 2023), VIGC (Huang et al., 2023) and KOSMOS-2 (Peng et al., 2023). Such initiatives mitigate the issue of hallucinations and lead to more reliable performances across visually grounded tasks, together with more rich multi-modality datasets (Zhao et al., 2023; He et al., 2023; Wang et al., 2023) resulting in the development of more advanced MLLMs (Zhang et al., 2023; Dong et al., 2024; Hong et al., 2023; Qi et al., 2023a;b). In a recent study, CoVLM (Li et al., 2023d) and V* (Wu & Xie, 2023) proposed to utilize a separate localization module to ground visual objects to enhance the performance of the LLM. In contrast, DualFocus is designed to enable the MLLM to ground a single question-relevant subregion encompassing all related objects, thereby imbuing the MLLM with a "question-grounding" capability.

## 2.3 HIGH RESOLUTION MLLMS

Recently, MLLMs primarily utilized fixed, lower-resolution inputs, typically 224 pixels (Liu et al., 2023b; Chen et al., 2023a; Zhu et al., 2023). LLaVA-1.5 (Liu et al., 2023a), and BLiVA (Hu et al., 2023) have sought to enhance performance by expanding input resolution to 336 pixels and integrating task-specific with global features, respectively. Moreover, advancements like Qwen-VL (Bai et al., 2023) have pushed resolution boundaries to 448 pixels and preserved original image sizes during inference, leading to more refined detail discernment. Notably, Monkey (Li et al., 2023f), (Li et al., 2023a), and Monkey (Li et al., 2023f) have significantly increased resolution with a high-resolution image divided into patches for details, accompanied by a low resolution for global information, introducing overwhelming question-irrelevant information and leading to quadratically increased computational cost. This paper introduces the DualFocus mechanism, which addresses the conflicting demands of micro-detail accuracy and macro-contextual understanding, providing a balanced solution for MLLM designs.

## 3 OUR APPROACH

In this section, we provide an initial overview of the Multi-modal Large Language Model (MLLM) (Sec. 3.1). Following that, we elucidate the methodology, covering aspects such as dataset construction (Sec. 3.2), the training phase (Sec. 3.3), and the inference process (Sec. 3.4).

## 3.1 PRELIMINARIES

The contemporary MLLMs usually adopt a modular architecture comprising a visual encoder $V$, a series of connection layers $W$, and a large language model $L$. Given an input image $v$ and its

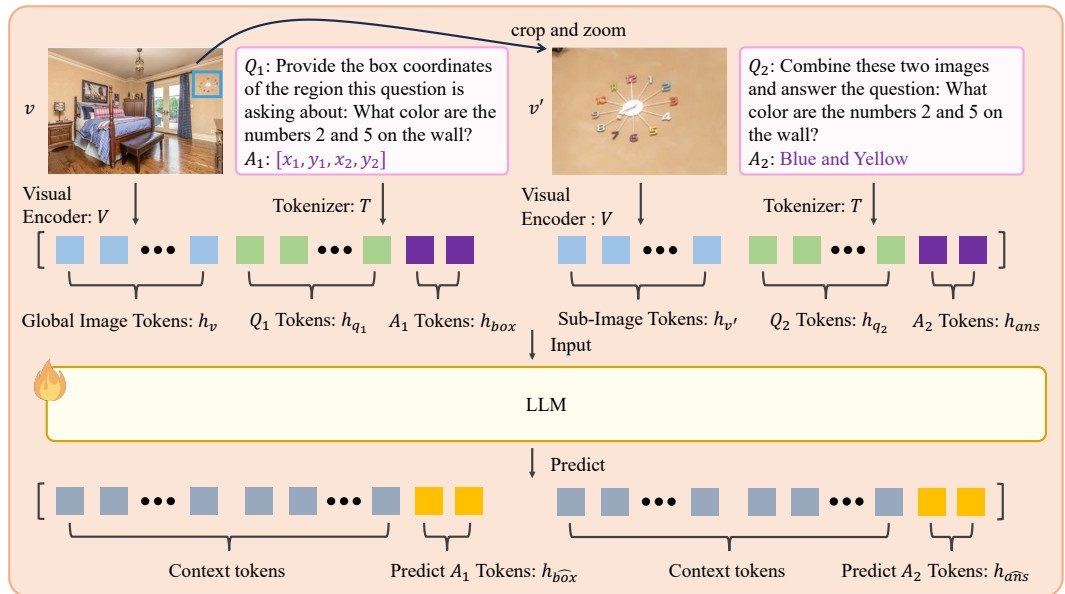

Figure 2: The **training framework** integrates two tasks into a single conversation for the LLM. First, the sub-region $v'$ is cropped and zoomed based on the target box $A_1$. The visual encoder $V$ processes both the global image $v$ and the sub-region $v'$ to extract visual tokens $h_v$ and $h_{v'}$. Simultaneously, the tokenizer $T$ converts the questions $Q_1$ and $Q_2$ into text tokens. These, along with the target tokens $h_{box}$, are concatenated and fed to the LLM, which makes a single forward pass to predict the target tokens.

corresponding question $q$, the visual encoder $V$ initially processes the image and encodes it into a set of visual tokens $z_v = V(v)$. These visual tokens are then transformed to align with the embedding space of the language model through the connection layers, such that $h_v = W(z_v)$. Concurrently, the text query $q$ is tokenized into linguistic tokens $h_q$ by the tokenizer $T$, becoming $h_q = T(x_q)$. These visual and text tokens are concatenated into a unified sequence $[h_v, h_q]$, which serves as the input to the decoder component of the large language model $L$. The model then utilizes this combined representation to infer the appropriate answer $ans = L([h_v, h_q])$, demonstrating the capability of these models to perform cross-modal reasoning and answer multimodal queries.

## 3.2 DATA CONSTRUCTION

To enhance the MLLM with the DualFocus mechanism, we curated a dataset derived from the extensive Visual Genome (VG) dataset (Krishna et al., 2017), which provides a diverse array of images coupled with corresponding questions, answers and annotated bounding boxes. These bounding boxes explicitly demarcate the regions of interest within the image pertinent to the question posed, potentially encompassing single or multiple objects, please refer to Appendix D for details.

**Ambiguity Filtration**. Initially, we scrutinize each data entry from VG to ensure its precision and clarity. During this process, we encountered instances where a question such as "What is the color of the person's shirt?" might correspond to a scene depicting multiple individuals, leading to ambiguity in the dataset. To establish a one-to-one mapping between visual cues and textual queries, we employed a strict filtering criterion to exclude such ambiguous samples. Through this rigorous refinement, we distilled our dataset to 143k unequivocal image-question pairs.

**Reformatting**. For enhanced interaction with our MLLM's training regime, we transmuted the dataset samples into a conversational format that encapsulates both the query and spatial awareness components. The schema of a data sample is as follows:

$Q_1$ : ``*Provide the coordinates of the region this question is asking about:* `<question>`

$A_1$ : `<box>`

$Q_2$ : `<sub img>`*Combine these two images and answer the question:* `<question>`

$A_2$ : `<answer>`

In the first round $(Q_1, A_1)$, we task the MLLM to deduce the important subregion `<box>` that is pertinent to the question `<question>` in the image ``, supplying it with micro details it needs to focus on. The subsequent round $(Q_2, A_2)$ is constructed to aggregate the augmented view `<sub img>` of the identified sub-region and the original contextualized image `` to infer the answer `<answer>`.

## 3.3 TRAINING

During training, we integrate our curated VG data with standard VQA datasets to enhance the model's capabilities on both micro and macro levels. We adhere to conventional MLLM training procedures using standard VQA datasets to equip the model with macro capabilities. Subsequent sections primarily focus on elaborating how we augment the model with the DualFocus mechanism through our transformed VG data. This enhancement is achieved by dividing the framework pipeline into two distinctive yet interconnected tasks as in Fig. 2, an efficient training implementation is adopted to train the two tasks in parallel.

**Task I: Grounding of the Question-Pertinent Subregion**. Given an image $v$ and the query $q$, we prompt the model with instruction $q_1$ to ground the region corresponding to the query $q$. To model this, we tokenize $q_1$ into tokens $h_{q_1}$ using the tokenizer $T(.)$, and the visual embedding $h_v$ is obtained from the input image $v$. The model prediction $\hat{box}$, representing the bounding box coordinates, is then inferred through the language model:

$$\hat{box} = L([h_v, h_{q_1}]), \tag{2}$$

where $\hat{box} = (\hat{x_1}, \hat{y_1}, \hat{x_2}, \hat{y_2})$, representing the coordinates of the two corners of the bounding box. The coordinates are expressed as numeric values embedded in natural language, with no additional formatting or special tokens, to maintain coherence with the LLM's language processing capabilities.

**Task II: In-depth Examination and Answer Generation**. Given the global image $v$ and the target sub-region $box$, we extract and upscale the sub-image $v'$ using the corresponding bounding box coordinates to maintain the original resolution: $v' = zoom(crop(v, box))$. To ensure that the context of the entire image is not lost, both the original image $v$ and the processed sub-image $v'$ are encoded by the same visual encoder $V$, producing two sets of visual tokens $h_v$ and $h'_v$, respectively. These visual tokens are concurrently concatenated with the text embedding generated from the first task, structured as $[h_v, h_{q_1}, h_{box}, h'_v, h_{q_2}]$. The model then employs this concatenated information to produce the final answer,

$$\hat{ans} = L([h_v, h_{q_1}, h_{box}, h_{v'}, h_{q_2}]). \tag{3}$$

**Objective Function**. The training loss is partitioned into two distinct segments corresponding to the abovementioned tasks. Since both the bounding box and the final answer are enunciated in natural language, we employ a standard cross-entropy loss function $\mathcal{L}_{CE}$ for each task. Formally, the collective loss is the aggregation of these binary components:

$$\mathcal{L}_{total} = \mathcal{L}_1(\hat{box}, box) + \mathcal{L}_2(\hat{ans}, ans), \tag{4}$$

where $\mathcal{L}_1$ computes the discrepancy between the actual ($box$) and predicted ($\hat{box}$) bounding boxes, and $\mathcal{L}_2$ quantifies the differential between the true final answer ($ans$) and the inferred one ($\hat{ans}$).

**Efficient Training Implementation**. For training efficiency, we offline crop the subregion images and integrate these two tasks into a single data sample with two-round conversations shown in format 1 . The model forwards once in next-token-prediction paradigm and is then optimized with the unified objective function (Equ. 4). The model gradually develops an adeptness in isolating and scrutinizing specific subregion within an image, thereby refining its capacity for fine-grained detail discernment.

## 3.4 INFERENCE

Upon training completion, our model acquires dual capabilities, namely the ability to generate macro-level answers ($\hat{ans}_{macro}$) directly from the holistic image and the capacity to produce micro-level answers ($\hat{ans}_{micro}$) using the fine-grained details from the predicted subregion. Thus, we adopt two distinct pathways for interpreting the given data.

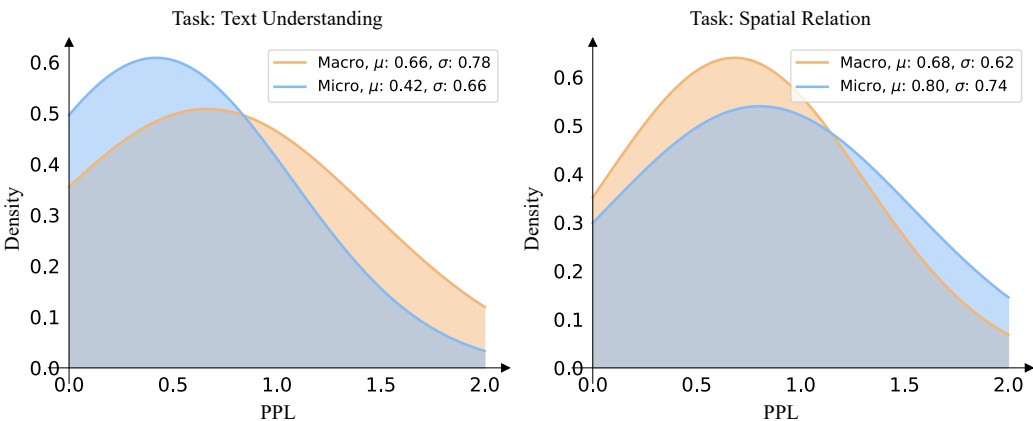

Figure 3: PPL distribution for Micro Answer compared to Macro Answer on tasks emphasizing different cognitive demands.

**Inference Pathways**. Specifically, the macro answer pathway, akin to the traditional method, which maintains the conventional inference process, directly generating an answer,

$$a\hat{n}s_{\text{macro}} = L([h_v, h_q]),\tag{5}$$

without emphasizing localized regions. Contrarily, the micro pathway mimics the training phase,

$$a\hat{n}s_{\text{micro}} = L([h_v, h_{q_1}, h_{\hat{box}}, h'_v, h_{q_2}]),\tag{6}$$

except during inference, we utilize the predicted bounding box $\hat{box}$ instead of the ground truth in Equ. 3. The micro pathway leverages the predicted bounding box $\hat{box}$ to focus on a specific sub-region.

**Perplexity-Guided Answer Selection**. To ascertain the most coherent response, we evaluate both $a\hat{n}s_{\text{macro}}$ and $a\hat{n}s_{\text{micro}}$ through their respective perplexity (PPL). The PPL serves as an estimate of the likelihood for a given sequence of tokens, with lower values indicating higher probability (better model confidence). This is given by:

$$\text{PPL}(a\hat{n}s) = \exp\left(-\frac{1}{N}\sum_{i=1}^{N}\log p(w_i|w_{<i})\right),\tag{7}$$

where $N$ is the number of tokens in the answer, $p(w_i|w_{<i})$ represents the model's estimated probability for token $w_i$ given the preceding context. The answer affiliated with the lower PPL is deemed more likely correct and thus selected as the final answer:

$$a\hat{n}s = \begin{cases} a\hat{n}s_{\text{macro}}, & \text{if } \text{PPL}(a\hat{n}s_{\text{macro}}) < \text{PPL}(a\hat{n}s_{\text{micro}}) \\ a\hat{n}s_{\text{micro}}, & \text{otherwise} \end{cases}\tag{8}$$

**The Motivation of Perplexity-Guided Answer Selection**. As depicted in Fig. 3 (Left), the micro answer demonstrates superior confidence ($\mu$) in scenarios requiring detailed discernment (*e.g.*, text understanding). However, its confidence degrades in tasks involving global comprehension (*e.g.*, spatial relationships), as shown in Fig. 3 (Right), despite the micro answer being generated by concatenating the original image and sub-image. We conjecture the degradation is due to the high dependence of the micro answer on the nearest image, *i.e.*, sub-image, akin to the high dependency observed in closely located text tokens. This motivates us to select the micro and macro answers via perplexity to integrate both perspectives during the inference. Using a perplexity-guided dual-path inference system, the MLLM dynamically switches between a global understanding and a focused comprehension dependent on the nature of the query, ultimately enhancing the model's efficacy.

Table 1: **Comparison with baseline methods on various benchmarks.** Our DualFocus consistently demonstrates improvements across various baselines and benchmarks.

| Method | Encoder-V | LLM | SEED$^{IMG}$ | MMB | GQA$^*$ | VQA$^T$ |
|---|---|---|---|---|---|---|
| LLaVA-1.5 **LLaVA-1.5 + DF** | ViT-L | Vicuna-7B | 66.2 **68.9 (+2.7)** | 64.3 **66.6 (+2.3)** | 67.2 **69.3 (+2.1)** | 58.2 **62.0 (+3.8)** |
| LLaVA-1.5 **LLaVA-1.5 + DF** | ViT-L | Vicuna-13B | 68.2 **71.0 (+2.8)** | 67.7 **70.7 (+3.0)** | 69.3 **74.5 (+5.2)** | 61.3 **65.5 (+4.2)** |
| Qwen-VL-Chat **Qwen-VL-Chat + DF** | ViT-G | Qwen-7B | 65.4 **66.6 (+1.2)** | 60.6 **63.2 (+2.6)** | 69.0 **73.0 (+4.0)** | 61.5 **63.7 (+2.2)** |

## 4 EXPERIMENTS

### 4.1 BENCHMARKS

To thoroughly assess DualFocus, we evaluated its performance across a spectrum of benchmarks, covering traditional academic Visual Question Answering (VQA) tasks (GQA (Hudson & Manning, 2019), TextVQA (Singh et al., 2019)) and recent benchmarks specifically designed for evaluating large multimodal models, namely MMBench (Liu et al., 2023c) and SEED (Li et al., 2023c). MMBench is constructed with manually designed questions to critically assess the model's vision-related reasoning and perceptual abilities. SEED, leveraging GPT-4 for generation, introduces a dataset of nearly 19,000 questions centered on images and videos. Herein, our emphasis is placed on the image component., referred to as SEED$^{IMG}$. GQA and TextVQA represent benchmarks in traditional Visual Question-answering tasks, with GQA assessing the model's ability to answer open-ended questions about images accurately and TextVQA focusing on questions requiring the understanding of text within images. Notably, GQA's evaluations revealed considerable variability due to discrepancies in the answer format. To address this, we employed GPT-3.5 to reformat answers into a multiple-choice question format, resulting in an adjusted benchmark referred to as GQA$^*$.

### 4.2 IMPLEMENTATION DETAILS

All experiments were performed using LLaVA-1.5 (Liu et al., 2023a) and Qwen-VL-Chat (Bai et al., 2023), adhering to their default hyper-parameters and training configurations unless stated otherwise. Our methodology uniquely altered the fine-tuning stage by incorporating the converted 143k VG data to fortify the MLLM with the DualFocus mechanism. For LLaVA-1.5, CLIP-ViT-L (Radford et al., 2021) served as the visual encoder at 336-resolution, and Vicuna 7B(13B) (Chiang et al., 2023) functioned as the LLM. During training we only freeze the visual encoder but fine-tune the connection layers and LLM. For Qwen-VL-Chat, CLIP-ViT-G was the visual encoder at 448 resolution, and Qwen-7B (Bai et al., 2023) functioned as its LLM. During training, given memory constraints, we freeze the visual encoder and LLM, only fine-tune the LoRA (Hu et al., 2022) and connection layers. The fine-tuning process for both models lasts a single epoch.

### 4.3 MAIN RESULTS

**Comparison with Baseline Model**. We first conducted comparisons against baseline MLLMs LLaVA-1.5 and Qwen-VL-Chat across four benchmarks: SEED, MMBench, GQA, and TextVQA. Our DualFocus mechanism notably enhances the performance of both methods, as outlined in Table 1. Specifically, our DualFocus improves LLaVA-1.5 with Vicuna-7B by 2.7, 2.3, 2.1, and 3.8, respectively. With the larger LLM, Vicuna-13B, DualFocus secures even more substantial gains: 2.8, 3.0, 5.2, and 4.2, on SEED, MMBench, GQA, and TextVQA, respectively. This trend is consistent when applying DualFocus to Qwen-VL-Chat, yielding boosts of 1.2, 2.6, 4.0 and 2.2 on the same benchmarks, respectively. These results highlight DualFocus's versatility and its capability to significantly elevate MLLM performance across diverse benchmarks.

**Comparison with SoTA Model.** Subsequently, we conduct a comparison of DualFocus with other SoTA MLLMs that vary in their input resolutions (Res), visual encoders (Encoder-V), and language models (LLM) on Table 2. We incorporate DualFocus into LLaVA-1.5 Vicuna-13B and

Table 2: **Comparison with SoTA methods on various benchmarks.** The best result and the second-best result should be indicated using bold and underlined, respectively.

| Method | Res | Encoder-V | LLM | SEED$^{IMG}$ | MMB | GQA$^*$ | VQA$^T$ |
|---|---|---|---|---|---|---|---|
| InstructBLIP | 224 | ViT-G | Vicuna-7B | 53.4 | 36.0 | - | 50.1 |
| LLaVA | 224 | ViT-L | Vicuna-7B | 25.5 | 34.1 | - | - |
| LLaVA-1.5 | 336 | ViT-L | Vicuna-7B | 66.2 | 64.3 | 67.2 | 58.2 |
| Share4V | 336 | ViT-L | Vicuna-7B | 69.7 | 68.8 | 70.5 | 60.4 |
| Qwen-VL-Chat | 448 | ViT-G | Qwen-7B | 65.4 | 58.2 | 69.0 | 61.5 |
| Monkey | 896 | ViT-G | Qwen-7B | 64.3 | 59.6 | - | **67.6** |
| OtterHD | 1024 | - | Fuyu-8B | - | 58.3 | - | - |
| BLIP-2 | 224 | ViT-L | Vicuna-13B | - | 46.4 | - | 42.5 |
| Shikra | 224 | ViT-L | Vicuna-13B | - | 58.8 | - | - |
| LLaVA-1.5 | 336 | ViT-L | Vicuna-13B | 68.2 | 67.7 | 69.3 | 61.3 |
| Share4V | 336 | ViT-L | Vicuna-13B | 70.8 | 68.5 | 71.1 | 62.2 |
| **LLaVA-1.5-DF (ours)** | 336 | ViT-L | Vicuna-13B | 71.0 | **70.7** | 74.5 | 65.5 |
| **Share4V-DF (ours)** | 336 | ViT-L | Vicuna-13B | **72.9** | **70.7** | **75.7** | 66.2 |

Table 3: Performance comparison on different inference strategies for baseline LLaVA-1.5 and our model. "Macro" and "Micro" refer to employ macro and micro answer pathways, respectively. "N/A" denotes the model failed to follow the instructions.

| Method | Macro | Micro | SEED$^{IMG}$ | VQA$^T$ |
|---|---|---|---|---|
| Base | ✓ | | 66.2 | 58.2 |
| | | ✓ | N/A | N/A |
| Ours | ✓ | | 66.7 | 58.6 |
| | | ✓ | 67.7 | 61.3 |
| | ✓ | ✓ | **68.9** | **62.0** |

ShareGPT4V (Chen et al., 2023b), a derivative of LLaVA, named as LLaVA-1.5-DF and Share4V-DF, exhibit superior performance across four benchmarks. Specifically, Share4V-DF surpasses its closest competitor by 2 on SEED. Similarly, LLaVA-1.5-DF leads the second-best performer by 1.9 on the MMBench. The results are even more pronounced on the GQA and Text-VQA benchmarks, which demand a higher capacity for detailed perception. Specifically, DualFocus improved Share4V by 4.7 and 4.0 on these benchmarks, respectively. While Monkey (Li et al., 2023f) achieves the highest 67.6 on TextVQA using a larger input of 896 x 896, it falls short on more comprehensive benchmarks like SEED and MMBench. In contrast, our Share4V-DF performs similarly on TextVQA with a much smaller input size of 336 x 336 and significantly better on the other two benchmarks, demonstrating DualFocus's ability to maintain a balance between a micro and macro perspective, making it a versatile and efficient mechanism for improving MLLM performance.

## 4.4 ABLATION STUDY

In this section, we first study impacts of each inference pathway and then explore the effect of each component and why they work. Unless otherwise specified, all ablations are based on LLaVA-1.5.

**Inference Pathway Analysis**. Table 3 illustrates the contributions of the micro and macro inference pathways to the performance. The initial results from the baseline model, LLaVA-1.5, indicate failure to implement the micro pathway due to the absence of training with similar directives. Integrating our custom 143k VG dataset enabled the model to follow the DF inference guidelines. However, this adaptation led to minor improvements, *i.e.*, increasing by +0.5 on SEED and +0.4 on TextVQA, suggesting that the dataset alone is insufficient to enhance performance.

However, the micro pathway results in a significant +1.0 gain on the SEED metric and a notable +2.7 gain on the TextVQA metric, supporting our hypothesis that the micro pathway excels in nuanced tasks. Conversely, global comprehension tasks benefit from the PPL selection, as evidenced by a

Table 4: Results on POPE. "LLaVA" refers to LLaVA-1.5. DualFocus is beneficial to mitigate Hallucination of MLLM. Here, A, P, and R denote adversarial, popular, and random split of POPE, respectively. "F1" and "Acc" denote F1 score and accuracy, respectively.

|  | F1(A) | Acc(A) | F1(P) | Acc(P) | F1(R) | Acc(R) |
|---|---|---|---|---|---|---|
| LLaVA | 84.2 | 85.2 | 86.2 | 87.3 | 87.4 | 88.2 |
| LLaVA + DF | **86.0** | **86.2** | **88.6** | **89.1** | **89.7** | **90.0** |

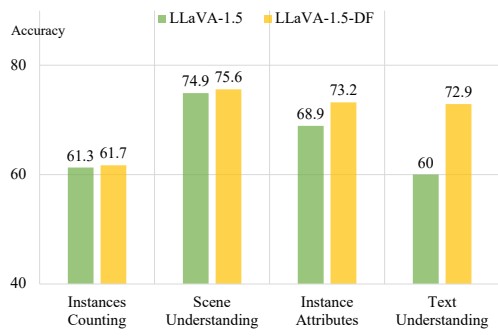

Figure 4: Accuracy of baseline LLaVA-1.5 and our LLaVA-1.5-DF on SEED Benchmark tasks across various granularities. Our Dual-Focus significantly improves accuracy on fine-grained tasks.

Table 5: Performance comparison of LLaVA-1.5, our LLaVA-1.5-DF, and permutation variants of LLaVA-1.5-DF on SEED, MMBench, GQA, and Text-VQA benchmark datasets. Here, 'Permute $x$ pixel' means we manually permute the predicted box of the sub-region of LLaVA-1.5-DF by $x$ pixels. Our DualFocus is robust to the permutation of sub-regions.

| Method | SEED | MMB | GQA | VQA$^T$ |
|---|---|---|---|---|
| LLaVA-1.5 | 66.2 | 64.3 | 67.2 | 58.2 |
| LLaVA-1.5-DF | **68.9** | **66.6** | **69.3** | **62.0** |
| Permute 10 pixel | 68.6 | 66.5 | 68.7 | 61.5 |
| Permute 25 pixel | 68.7 | 66.3 | 68.6 | 61.3 |
| Permute 150 pixel | 67.1 | 65.2 | 67.7 | 59.0 |

+1.2 gain on the SEED metric and a moderate +0.7 gain on the TextVQA metric. This underscores the importance of employing the appropriate inference pathway based on the task's requirement.

**Hallucination Mitigation**. Hallucination within MLLM presents a critical challenge where the model creates imaginary content that is not present in the image. The benchmark POPE (Li et al., 2023e) is designed to evaluate such hallucinations in MLLM through three distinct data splits: adversarial (A), popular (P), and random (R). As indicated in Table 4, integrating our DualFocus into MLLM yields substantial improvements in accuracy and the F1 score across these data splits. Specifically, it improves baseline by 2.4 and 2.3 on the F1 score of splits "P" and "R", respectively. Even on the most difficult split "A", it yields 1.8 gains on the F1 score. The effectiveness of DF is attributed to the fact that our DualFocus directs the model's attention toward specific, relevant parts of an image in connection to the posed question, reducing the generation of non-pertinent features and subsequently diminishing the likelihood of hallucinations.

**Fine-Grained Perception Enhancement**. In this section, we delve into the effectiveness of the DualFocus mechanism on tasks emphasizing different cognitive demands. We use the SEED benchmark because it provides a comprehensive assessment of a model's capabilities across different dimensions and levels of detail. Specifically, we examine four key dimensions: Instance Counting, Scene Understanding, Instance Attributes, and Text Understanding. The first two dimensions primarily concern the broader context of a situation, emphasizing a macro perspective. In contrast, the latter two focus on more intricate, micro-level details. Experiment results are presented in Figure 4, illustrating that while our DualFocus mechanism delivers modest improvements in the domains of instance counting and Scene Understanding (+0.4, +0.7), it significantly enhances performance on Instance Attributes and Text Understanding (+4.3, +12.9). These results underscore the effectiveness of the DualFocus approach, particularly in tasks requiring acute attention to detail, thereby confirming its utility in dissecting and interpreting finer elements within data.

**Robustness of Question-Pertinent Sub-region Grounding**. DualFocus is learned to ground the sub-region pertinent to the question rather than a single object, as shown in Fig. 5. The sub-region aims to encompass all related objects, though it may not be highly precise, the final answer is robust to the permutation of the sub-region, detailed in Tab. 5. We randomly permute the predicted sub-region by different pixels. DualFocus is robust to small pixel permutations (10, 25 pixels). Even with a large permutation (150 pixels), thanks to our framework that integrates both the micro-view (sub-region) and the macro-view (the global image), DualFocus still ensures gains over the baseline.

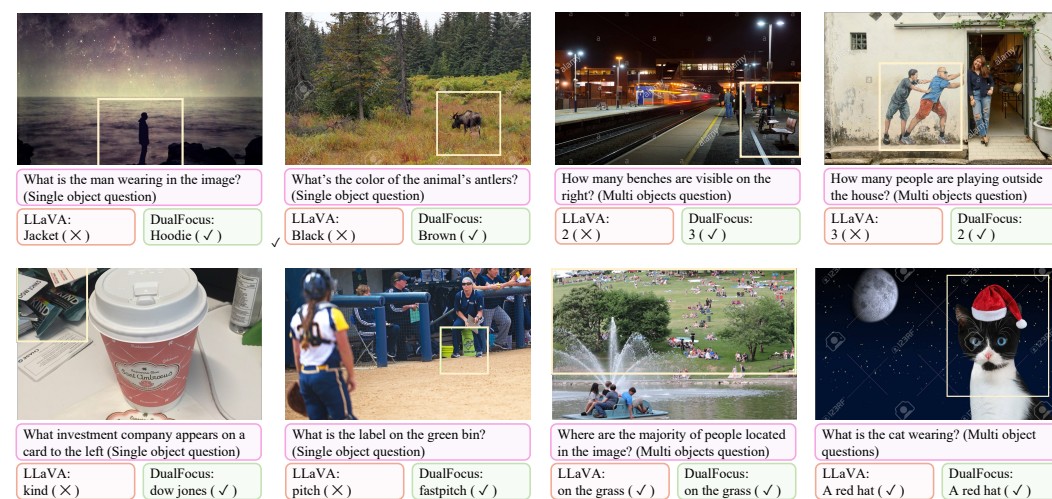

Figure 5: Comparative visualizations between LLaVA-1.5 and DualFocus on different VQA scenarios. In the scenario of a single object question, LLaVA-1.5 often struggles to capture micro details. In contrast, our DualFocus mechanism leverages the zoomed-in question-related sub-region (highlighted with a yellow bounding box) to achieve improved discernment of fine-grained details. In the scenario of the multi-object question, our DualFocus is versatile to ground all objects pertinent to the question and then accurately answer the question.

Table 6: Comparison between DualFocus with high-resolution MLLM LLaVA-NeXT (Liu et al., 2024). Dual-Focus strikes a better trade-off between efficiency and performance. Additionally, integrating DualFocus with LLaVA-HighRes can significantly enhance its overall performance. The inference time is measured using a single A100 GPU on the MMBench benchmark.

| Model (7b) | SEED | MMB | GQA | TextVQA | Time (ms) |
|---|---|---|---|---|---|
| LLaVA-1.5 | 66.2 | 64.3 | 67.2 | 58.2 | 117 |
| LLaVA-1.5-DF | 68.9 | 66.6 | 69.3 | 62.0 | 245 |
| LLaVA-HighRes | 68.1 | 65.4 | 68.0 | 63.1 | 443 |
| LLaVA-HighRes-DF | **69.5** | **66.9** | **71.6** | **63.7** | **622** |

**Integration with High-Resolution MLLMs** We compare DualFocus with another high-resolution MLLM, LLaVA-NeXT (Liu et al., 2024). LLaVA-NeXT processes high-resolution images by dividing them into up to four low-resolution patches of 336 pixels, along with the original global patch. These patches are encoded by the same visual encoder, resulting in many image tokens. In contrast, DualFocus focuses on a single, relevant sub-region of the image related to the question, producing significantly fewer tokens. We present a comparison of LLaVA-HighRes and DualFocus in Table 6 . Since LLaVA-NeXT does not release its training data, we trained it on the same dataset as DualFocus and named it LLaVA-HighRes for a fair evaluation. DualFocus outperforms LLaVA-HighRes on SEED, MMB, and GQA benchmarks. While it slightly trails behind on Text-VQA, it achieves significantly faster inference speeds—245 ms/iteration compared to LLaVA-HighRes's 443 ms/iteration. This indicates that DualFocus strikes a good balance between performance and efficiency. For more details on the inference framework, please see Sec. A. Additionally, DualFocus can be integrated with LLaVA-HighRes by including the sub-region identified by the model as a fifth local patch. This integration significantly enhances LLaVA-HighRes's performance across all four benchmarks, particularly yielding a notable 3.6-point improvement on the GQA benchmark.

## 5 CONCLUSION

In this work, we introduced DualFocus, a novel approach to enhance the performance of Multi-modal Large Language Models (MLLMs) by integrating both macro and micro perspectives for improved visual question answering. Through comparative studies, DualFocus demonstrated superior capability in handling detailed features and mitigating hallucination, thereby outperforming existing methods. This method not only advances MLLM efficacy but also paves the way for more human-like visual reasoning in AI.

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

# A  INFERENCE FRAMEWORK

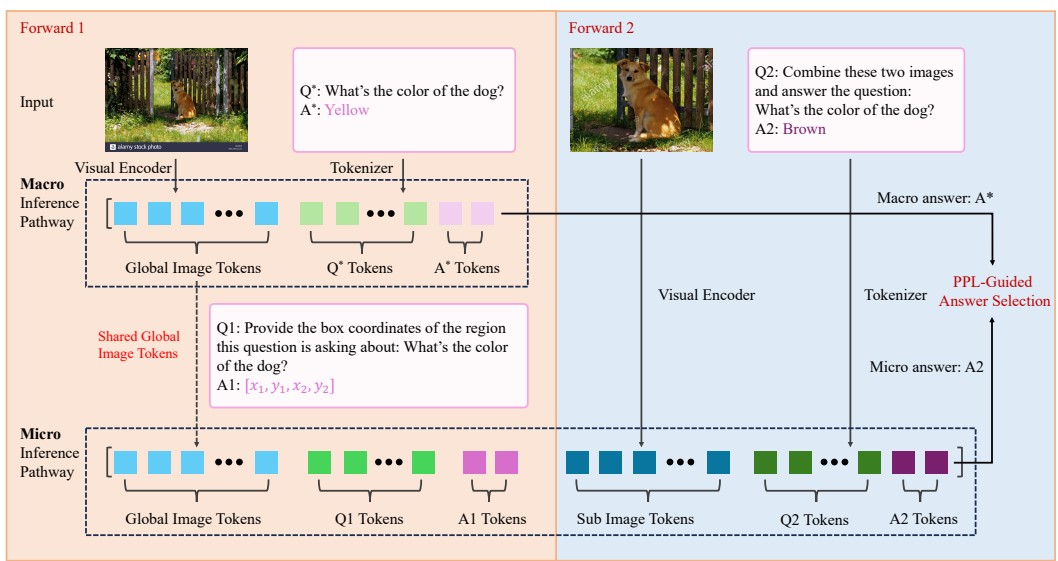

Figure 6: Illustration of the two-round inference framework in DualFocus. The first forward pass performs both macro inference (Q*, A*) and the first stage of micro inference (Q1, A1) to obtain the macro answer A* and the predicted box A1. The second forward pass (Q2, A2) performs the second stage of the micro inference to obtain the micro answer A2. The KV cache of the global image tokens and the Q1 tokens are reused for efficiency. The final answer is selected using PPL for the best macro and micro answers.

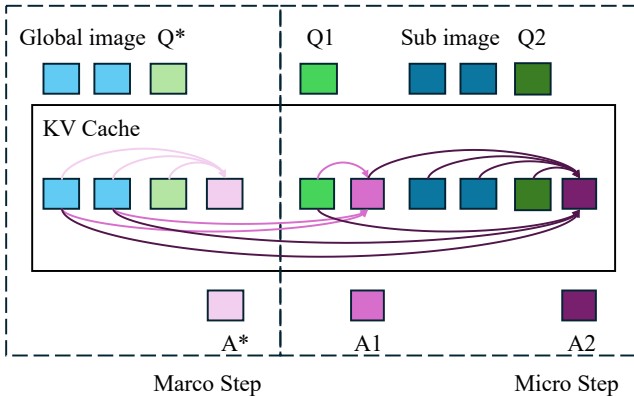

Figure 7: Demonstration of KV cache usage during inference. The arrows at the bottom represent the reuse of the pre-computed KV cache from the previous stage. During A1 prediction, the KV cache of the global image is reused. For A2 prediction, the KV cache of both the global image and Q1 tokens is reused.

Table 7: Comparison of inference time and GPU memory between LLaVA-1.5 and our LLaVA-1.5-DF. Both metrics were measured using a single A100 GPU.

| Model | Inference Time | GPU Memory |
|---|---|---|
| LLaVA-1.5 | 117 ms | 17.0 GB |
| LLaVA-1.5-DF | 245 ms | 18.6 GB |

Figure 6 shows the detailed implementation of the two-round inference framework for our Dual-Focus. The inference procedure entails two forward passes. In the first forward pass, both macro inference (Q*, A*) and the first stage of micro inference (Q1, A1) are performed using the same

global image. This approach allows for the reuse of global image tokens and their KV cache during the first stage of micro inference.

In the second pass (Q2, A2), the second stage of micro inference is executed. The pre-computed KV cache of global image tokens and Q1 tokens from the first pass are reused, improving efficiency in generating the answer tokens, A2. This optimized approach ensures that DualFocus inference time is approximately twice as fast as the baseline, as shown in Table 7.

Finally, we use PPL to select the best answer among the macro and micro answers.

## B    PERPLEXITY-GUIDED ANSWER SELECTION

Table 8: Performance comparison of different methods for implementing the PPL strategy on MMBench. "LLaVA" and "Qwen" refer to LLaVA-1.5 and Qwen-VL-Chat. "LLaVA + PPL" denotes using PPL to choose answers generated by LLaVA with two distinct prompts, a process mirrored in "Qwen + PPL". "LLaVA + Qwen + PPL" refers to using PPL to select the best answer from LLaVA and Qwen.

| Method | LLaVA | Qwen | PPL | Acc |
|--------|:-----:|:----:|:---:|:----:|
|        | ✓     |      |     | 64.3 |
|        | ✓     |      | ✓   | 64.1 |
| Base   |       | ✓    |     | 60.6 |
|        |       | ✓    | ✓   | 60.7 |
|        | ✓     | ✓    | ✓   | 62.5 |
| Our    | LLaVA-DF |  |     | 66.6 |
|        | Qwen-DF  |  |     | 63.2 |

During inference, we utilized Perplexity (PPL) to select answers by combining the micro and macro inference pathways, essentially creating a unique assembly method. We further examine various assembly approaches, detailed in Table 8. The first strategy involves using PPL to merge answers from the same models but varying input formats, labeled as "LLaVA + PPL" and "Qwen + PPL". Given that the base model is limited to macro inference pathways, we applied two distinct prompts. Results indicate a minor impact on performance, with changes of -0.2 and +0.1, respectively. Another assembly strategy involves using PPL to merge answers from different models, tagged as "LLaVA + Qwen + PPL". This approach significantly improved Qwen by +1.9, yet it reduced LLaVA's performance by 1.8. We suspect this variance results from differing model architectures and training methodologies. In contrast, DualFocus integrates micro and macro pathways within each model, applied to LLaVA and Qwen, resulting in substantial gains of +2.3 and +2.6, respectively, higher than "LLaVA + Qwen + PPL". This suggests combining micro and macro inferences within a single model outperforms assembling answers across different models.

## C    PERPLEXITY-GUIDED ADAPTIVE INFERENCE.

DualFocus selects the best answer via PPL-guided answer selection. Figure 3 shows that the PPL value correlates with confidence in correct answers. This metric helps determine whether to do micro inference or not. We conducted experiments on the GQA benchmark to skip micro-inference when the PPL value falls below a threshold, as shown in Table 9. DualFocus operates with a PPL threshold 0, and micro-inference is applied uniformly to all samples. At a threshold of 0.7, 22% of questions bypass micro-inference with only a 0.1% performance drop. At a threshold of 0.8, 58.1% of questions bypass micro-inference with only a 0.5% performance drop, indicating that micro-inference is unnecessary when the PPL value of the macro answer is below a certain threshold. Higher thresholds result in faster inference but lower performance.

Table 9: Balancing inference time and accuracy using a PPL-guided adaptive inference strategy. The micro-inference pathway is bypassed when the PPL value falls below a specific threshold, leading to faster inference with minimal impact on performance.

| PPL Threshold | Num Skip (%) | GQA Accu | Inference Time (ms) |
|---|---|---|---|
| 0 | 0 | 69.3 | 245 |
| 0.7 | 2756 (22.0%) | 69.2 | 194 |
| 0.75 | 5020 (40.0%) | 69.0 | 182 |
| 0.8 | 7312 (58.1%) | 68.8 | 155 |
| 0.85 | 9519 (75.7%) | 68.2 | 140 |
| ∞ | 12578 (100.0%) | 67.1 | 117 |
| LLaVA-1.5 baseline | - | 67.2 | 117 |

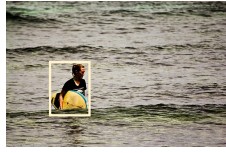
$Q_1$: Please provide the bounding box coordinate of the region this question asks: What is the women holding?
$A_1$: [0.20, 0.41, 0.37, 0.74]

$Q_2$: Combine these two images and answer the question: What is the women holding?
$A_2$: A surfboard.

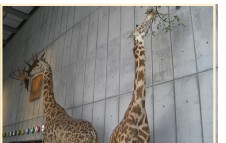
$Q_1$: Please provide the bounding box coordinate of the region this question asks: What is wall made of?
$A_1$: [0.01, 0.00, 1.00, 1.00]

$Q_2$: Combine these two images and answer the question: What is wall made of?
$A_2$: Concrete.

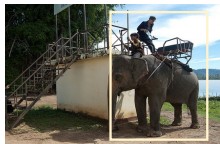
$Q_1$: Please provide the bounding box coordinate of the region this question asks: What is photo of?
$A_1$: [0.00, 0.00, 1.00, 1.00]

$Q_2$: Combine these two images and answer the question: What is photo of?
$A_2$: Large ground with animals.

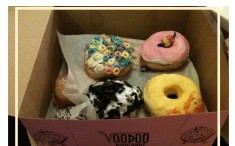
$Q_1$: Please provide the bounding box coordinate of the region this question asks: What is in the box?
$A_1$: [0.03, 0.04, 0.94, 0.92]

$Q_2$: Combine these two images and answer the question: What is in the box?
$A_2$: Sweet doughnut treats.

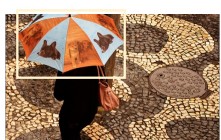
$Q_1$: Please provide the bounding box coordinate of the region this question asks: What are the people doing?
$A_1$: [0.52, 0.06, 0.93, 0.93]

$Q_2$: Combine these two images and answer the question: What are the people doing?
$A_2$: Loading an elephant.

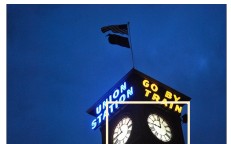
$Q_1$: Please provide the bounding box coordinate of the region this question asks: What is the quality of the back of the both clocks?
$A_1$: [0.46, 0.73, 0.78, 0.99]

$Q_2$: Combine these two images and answer the question: What is the quality of the back of the both clocks?
$A_2$: Glowing.

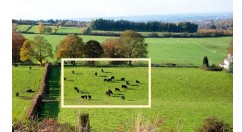
$Q_1$: Please provide the bounding box coordinate of the region this question asks: Why are there dogs on the umbrella?
$A_1$: [0.05, 0.01, 0.56, 0.42]

$Q_2$: Combine these two images and answer the question: Why are there dogs on the umbrella?
$A_2$: Decoration.

$Q_1$: Please provide the bounding box coordinate of the region this question asks: What kind of animals are pictured?
$A_1$: [0.25, 0.51, 0.61, 0.71]

$Q_2$: Combine these two images and answer the question: What kind of animals are pictured?
$A_2$: Cows.

Figure 8: Examples of our curated VG dataset.

Table 10: Impact of additional VG samples on performance.

| Model (7B) | SEED | MMBench | GQA | TextVQA |
|---|---|---|---|---|
| LLaVA-1.5 | 66.2 | 64.3 | 67.2 | 58.2 |
| LLaVA -1.5 + Our VG | 66.7 | 64.4 | 67.1 | 58.6 |
| DualFocus | **68.9** | **66.6** | **69.3** | **62.0** |

## D CURATED VG DATASET

**Detailed Statistic Analysis of Our Curated Dataset**. Our dataset comprises a total of 7,825 images. It is worth noting that each image may be associated with multiple questions, culminating in 143,978 data entries. It should be highlighted that the original LLaVA training database encompasses approximately 86,000 VG (Visual Genome) data points, and we statistic that 1,701 images present within our dataset do not feature in the LLaVA training data, which contributes minimally to additional knowledge. Table 10 shows that including our VG samples has negligible performance gains for the LLaVA-1.5 baseline, indicating the gains mainly benefit from the DualFocus paradigm.

**Detailed Examples of Our Curated Dataset**. Fig. 8 demonstrates some examples from our dataset. As mentioned in the data format 1, each data entry is systematically organized as a two-round conversation, accompanied by the original image and the subregion image (highlighted with the yellow bounding box).

It merits emphasis that the sub-region may contain single or multiple objects pertinent to the posed question. When the query necessitates a broader contextual comprehension, *e.g.*, the first example in the second row asks "What is photo of", the subregion will approximate the whole image. This design is fundamental to ensuring the model is adept at identifying and grounding in the question-relevant subregions.

## E    DIFFERENT GROUNDING STRATEGY

Table 11: Results for different sub-region grounding methods. DualFocus-DINO refers to replacing the sub-region identified by the MLLM with the region identified by GroundingDINO. DualFocus consistently outperforms DualFocus-DINO on both the SEED and MMBench benchmarks.

| Model (7B) | SEED | MMBench |
|---|---|---|
| LLaVA-1.5 | 66.2 | 64.3 |
| DualFocus-DINO | 68.1 | 65.3 |
| DualFocus | **68.9** | **66.6** |

DualFocus identifies a single sub-region that includes all relevant objects related to the query. What happens when we replace this region with other grounding methods? Table 11 presents results from substituting the MLLM's identified sub-region with that of GroundingDINO. While introducing region information improves performance, DualFocus-DINO achieves gains of 1.9 and 1.0 on the SEED and MMBench benchmarks, respectively. However, it still falls short compared to DualFocus, which leverages the MLLM's comprehensive question understanding to identify a more effective sub-region.

## F    LIMITATION

While DualFocus demonstrates significant advancements in multi-modal large language models (MLLMs), there are some inherent limitations of MLLMs. First, MLLMs often demand substantial computational resources, making them less accessible for researchers and practitioners with limited infrastructure. Second, the rapid evolution of modalities and data types presents an ongoing challenge in maintaining and updating MLLMs to keep pace with the latest developments. Additionally, MLLMs are susceptible to biases in their training data, potentially perpetuating and amplifying these biases in their outputs.

## G    BROADER IMPACT

The advent of DualFocus, an innovative approach in multi-modal large language models (MLLMs), heralds significant societal implications, encompassing both beneficial and adverse facets.

On the positive side, DualFocus is poised to enhance machine comprehension of visual and textual data, broadening the horizons of applications in assistive technologies, education, and information retrieval. Specifically, its nuanced understanding could revolutionize how visually impaired individuals interact with digital content, enabling these technologies to provide more accurate and contextually relevant information. Furthermore, in educational settings, this advanced comprehension capability can facilitate personalized learning experiences, particularly in visually intensive subjects such as biology and geometry, by adapting content to cater to the learner's inquiry with precision.

On the negative side, the potential for misuse in surveillance and data privacy cannot be overlooked. The ability of DualFocus models to interpret visual data with granular detail might pave the way for intrusive surveillance practices, risking individuals' privacy and autonomy.

In summary, while DualFocus promises to unlock new frontiers in human-computer interaction, echoing the dual nature of technological progress, it necessitates rigorous ethical scrutiny and equitable access strategies to ensure its benefits are universally accessible, mitigating societal risks.

