# OpenReview forum: "DualFocus: Integrating Macro and Micro Perspectives in Multi-modal Large Language Models"
_ICLR.cc/2025/Conference — ICLR 2025 Conference Withdrawn Submission_

### Official Review · Reviewer_y1vQ · 2024-10-30

**Soundness:** 2
**Presentation:** 2
**Contribution:** 2
**Rating:** 3
**Confidence:** 5

**Summary:**

The paper introduces DualFocus, which addresses questions related to images (primarily high-resolution) from both global and regional perspectives. It systematically selects responses through PPL. Improvements have been achieved on several perception benchmarks.

**Strengths:**

This paper:
1. proposes a feasible (and potentially applicable) solution for high-resolution QA problems. It utilizes the model’s region perception ability to dynamically sample image crops and provide further answers, enhancing detail perception without significantly increasing image tokens.
2. explicitly supervises region prediction, effectively improving performance.
3. uses PPL (Perplexity) to dynamically select answers, effectively balancing responses from both perspectives.

**Weaknesses:**

1. Regarding motivation, there are two core issues:
- While cropping out regions as independent contexts can help the model better perceive these areas, it may weaken the model’s perception of other local areas at that moment. This seems to conflict with human perceptual mechanisms.
- Methods like Qwen2VL, which use a sufficient number of tokens to encode each image, are becoming mainstream and are engineering-wise feasible. Given this trend, is it still necessary to employ such dynamic region sampling methods?
2. Technically, this paper employs some trivial techniques to improve upon V*, achieving certain effects but lacking in novelty and insight.
3. This paper leverages the model’s own capability to perform region search, replacing the heatmap-based search algorithm used by V*. In fact, grounding methods based on MLLM are already quite mature. For instance, Shikra/QwenVL directly model bounding boxes, and GPT-4ROI incorporates regions into context. While this article claims to model this capability, it lacks a reasonable evaluation and an assessment of how this modeling approach impacts general performance.

**Questions:**

See weaknesses

---

### Official Review · Reviewer_jNhJ · 2024-11-04

**Soundness:** 2
**Presentation:** 3
**Contribution:** 2
**Rating:** 5
**Confidence:** 4

**Summary:**

The paper introduces DualFocus, a novel approach for multi-modal large language models (MLLMs) that integrates macro (global) and micro (detailed) perspectives to enhance performance on vision-language tasks. Inspired by human perceptual behavior, DualFocus allows the model to examine an entire image globally and then focus on specific sub-regions for detailed analysis, emulating a zoom-in function. This dual analysis is achieved by generating both a macro-level answer and a micro-level answer, with a perplexity (PPL)-guided selection process choosing the more confident response as the final answer.

**Strengths:**

The paper is well-written and easy to understand.

The DualFocus strategy is an interesting and novel approach, integrating macro and micro perspectives to improve visual comprehension.

The effectiveness of this approach, especially when combined with the perplexity (PPL) metric for answer selection, is well-supported by the experimental results provided in the paper.

**Weaknesses:**

1. **Differentiation from Prior Work**: The paper lacks a clear discussion on how DualFocus fundamentally differs from similar prior approaches, such as the V* model. Clarifying the unique contributions of DualFocus in comparison to existing methods would strengthen the case for its novelty.

2. **Performance Gains from Ppl-Guided Answer Selection:** Perplexity-guided answer selection is a commonly used strategy to boost performance during inference. It is unclear whether the observed performance gains can be attributed specifically to DualFocus’s capabilities in enhancing perceptual understanding, as claimed by the authors, or if they are largely due to the PPL metric. Testing this strategy with a baseline model (e.g., LLaVA-1.5) under similar dual inference conditions could provide a clearer understanding of where these gains originate. **This also makes me question the contribution of this paper.**

3. **Further Comparison**: The comparisons in the paper could be more comprehensive. If we use an external detection model to generate bounding boxes as prompts for baseline models, can we also yield performance improvements?

**Questions:**

See weaknesses.

---

### Official Review · Reviewer_mMDH · 2024-11-06

**Soundness:** 3
**Presentation:** 3
**Contribution:** 3
**Rating:** 6
**Confidence:** 3

**Summary:**

This paper presents DualFocus, an innovative approach designed to improve multi-modal large language models (MLLMs) in visual question answering by integrating both macro (global) and micro (focused) perspectives. Inspired by human perceptual behaviors, DualFocus enables models to examine an entire image first and then zoom in on question-relevant regions for finer detail. This dual-path approach allows for accurate responses to questions requiring both broad and detailed visual understanding. Through extensive experiments on benchmarks like SEED and GQA, the authors demonstrate that DualFocus enhances model accuracy and reduces hallucinations, addressing significant limitations in current MLLMs that struggle with fine-grained details.

**Strengths:**

1. Innovative Approach: DualFocus presents a novel approach by combining macro and micro perspectives, offering a unique method for MLLMs to enhance their visual question answering capabilities.
2. Comprehensive Evaluation: The paper thoroughly evaluates DualFocus across multiple benchmarks, including SEED, MMBench, and TextVQA, showing its versatility and effectiveness in improving MLLM performance.
3. Practical Impact: By reducing hallucinations and improving the model’s ability to capture fine-grained details, DualFocus addresses practical challenges in MLLMs, making it potentially impactful for real-world applications that require accurate visual interpretation.

**Weaknesses:**

1. Dataset Limitation: The reliance on the Visual Genome dataset for subregion identification may introduce limitations in generalizability, as VG may not represent general real-world visual contexts.
2. Computational Cost: DualFocus introduces additional computational overhead with its two-step zoom and evaluation process, which may limit further research and its deployment in resource-constrained environments.
3. Limited Analysis of Failure Cases: While the paper demonstrates the model's improvements, there is limited discussion on scenarios where DualFocus might fail, especially for highly ambiguous or complex images where even zoomed views could mislead the model.

**Questions:**

1. Dataset Diversity: Given the potential limitations of the Visual Genome dataset, did the authors consider constructing a generalized sub-region grounding dataset to improve the model's robustness across diverse visual contexts?
2. Computational Efficiency: Since DualFocus introduces additional computational layers, have the authors explored lightweight optimizations or alternative methods to achieve similar performance enhancements with reduced resource requirements?
3. Failure Case Analysis: Could the authors provide further insights into cases where DualFocus may not perform as expected? A more detailed analysis of such scenarios would be valuable for refining the model and understanding its limitations in practical applications.
4. Lack of Analysis on Visual Encoder: The ViT visual encoder is only pretrained on holistic images, whereas in this paper, ViT is used for both holistic images and sub-region zoomed images. This may create an input domain gap, making it challenging for ViT to accurately represent sub-region image features. Could the authors address this issue?

---

### Note · Authors · 2024-11-15

I have read and agree with the venue's withdrawal policy on behalf of myself and my co-authors.